# Separation and Identification of Antioxidants and Aldose Reductase Inhibitors in *Lepechinia meyenii* (Walp.) Epling

**DOI:** 10.3390/plants10122773

**Published:** 2021-12-15

**Authors:** Guanglei Zuo, Kang-Hoon Je, Yanymee N. Guillen Quispe, Kyong-Oh Shin, Hyun Yong Kim, Kang Hyuk Kim, Paul H. Gonzales Arce, Soon Sung Lim

**Affiliations:** 1Department of Food Science and Nutrition, Hallym University, 1 Hallymdeahak-gil, Chuncheon 24252, Korea; guangleizuo@foxmail.com (G.Z.); tlsruddhek@hallym.ac.kr (K.-O.S.); khy9514@nate.com (H.Y.K.); M20023@hallym.ac.kr (K.H.K.); 2Institute of Korean Nutrition, Hallym University, 1 Hallymdeahak-gil, Chuncheon 24252, Korea; jekh@hallym.ac.kr; 3Department of Molecular Medicine and Biopharmaceutical Sciences, Graduate School of Convergence Science and Technology, Seoul National University, Seoul 151742, Korea; yany24@snu.ac.kr; 4Laboratorio de Florística, Departamento de Dicotiledóneas, Museo de Historia Natural—Universidad Nacional Mayor de San Marcos, Avenida Arenales 1256, Lima 14-0434, Peru; pgonzalesarce@hotmail.com; 5Institute of Natural Medicine, Hallym University, 1 Hallymdeahak-gil, Chuncheon 24252, Korea

**Keywords:** *Lepechinia meyenii* (Walp.) Epling, antioxidant, aldose reductase inhibitor, high-speed counter-current chromatography, rosmarinic acid, quantification

## Abstract

We previously reported that *Lepechinia meyenii* (Walp.) Epling has antioxidant and aldose reductase (AR) inhibitory activities. In this study, *L. meyenii* was extracted in a 50% MeOH and CH_2_Cl_2_/MeOH system. The active extracts of MeOH and 50% MeOH were subjected to fractionation, followed by separation using high-speed counter-current chromatography (HSCCC) and preparative HPLC. Separation and identification revealed the presence of caffeic acid, hesperidin, rosmarinic acid, diosmin, methyl rosmarinate, diosmetin, and butyl rosmarinate. Of these, rosmarinic acid, methyl rosmarinate, and butyl rosmarinate possessed remarkable antioxidant and AR inhibitory activities. The other compounds were less active. In particular, rosmarinic acid is the key contributor to the antioxidant and AR inhibitory activities of *L. meyenii*; it is rich in the MeOH extract (333.84 mg/g) and 50% MeOH extract (135.41 mg/g) of *L. meyenii* and is especially abundant in the EtOAc and *n*-BuOH fractions (373.71–804.07 mg/g) of the MeOH and 50% MeOH extracts. The results clarified the basis of antioxidant and AR inhibitory activity of *L. meyenii,* adding scientific evidence supporting its traditional use as an anti-diabetic herbal medicine. The HSCCC separation method established in this study can be used for the preparative separation of rosmarinic acid from natural products.

## 1. Introduction

Diabetes mellitus, characterized by hyperglycemia and diabetic complications, is one of the most common chronic degenerative diseases worldwide, with nearly 463 million cases reported in 2019 alone [1]. Multi-therapeutical strategies beyond glycemic control are required to treat diabetes and its complications. Among these, aldose reductase (AR) and oxidative stress are considered significant therapeutic targets [2,3]. AR is the key enzyme in the polyol pathway that catalyzes NADPH-dependent reduction of glucose to sorbitol [3]. In hyperglycemic conditions, AR is activated and the polyol pathway flux is increased; it causes depletion of NADPH and overproduction of sorbitol, leading to cellular oxidative stress and sorbitol-induced osmotic stress, which are implicated in diabetic complications in insulin-independent tissues, including kidney, lens, retina, and neural tissues [3,4]. Moreover, reactive oxygen species (ROS) and the resulting oxidative stress are key contributors to diabetic complications [5,6]. Therefore, inhibition of AR and ROS/oxidative stress is considered a therapeutic target for treating diabetic complications [2,3]. 

Natural products are important resources of anti-diabetic agents, among which herb medicines play a significant role. *Lepechinia meyenii* (Walp.) Epling (*L. meyenii*), belonging to the Lamiaceae family, is native to Argentina, Bolivia, and Peru [7]. The infusion of *L. meyenii* is used as a traditional herb medicine in Peru to treat diabetes, cough, inflammation, diarrhea, spasm, burning sensation in the stomach, and pain in the stomach and joints [8,9,10]. In our ongoing research to screen and isolate potential anti-diabetic agents from natural products [11,12,13,14], we found that the 70% MeOH extract of *L. meyenii* has strong antioxidant and AR inhibitory activities [15], indicating the presence of potent antioxidants and AR inhibitors in this plant. Many diterpenoids have been previously identified from *L. meyenii*. Recently, it has been reported to have antibacterial and tyrosinase inhibitory activities; therefore, carnosol, rosmanol, carnosic acid, *p*-coumaric acid, caffeic acid, and rosmarinic acid were isolated and identified [16,17]. Nevertheless, the antioxidant and AR inhibitory compounds in *L. meyenii* remain unidentified, which prompted us to separate and identify the underlying bioactive compounds from this plant.

High-speed counter-current chromatography (HSCCC) is a liquid–liquid partition-based chromatography widely used in separating natural products [18,19,20,21]. It has advantages of solid support-free, high sample-loading capacity, no irreversible adsorption, and low risk of sample denaturation [22], and, therefore, was used for preparative separation of the antioxidants and AR inhibitors from *L. meyenii* in this study. Samples with simple component composition and low quantity were separated using preparative HPLC (pre-HPLC). Moreover, because rosmarinic acid, methyl rosmarinate, and butyl rosmarinate separated from *L. meyenii* in this study possessed remarkable antioxidant and AR inhibitory activities, ethyl and propyl rosmarinates were further synthesized to study the esterification effects of rosmarinic acid using short-chain primary alcohol (≤C4) on its antioxidant and AR inhibitory activities. 

Therefore, we aimed to separate and identify the antioxidants and AR inhibitors in *L. meyenii.* In addition, the esterification effects of rosmarinic acid on its antioxidant and AR inhibitory activities using short-chain primary alcohols (≤C4) were examined, and the major compound, rosmarinic acid, was quantified in all the extracts and fractions of *L. meyenii*.

## 2. Results

### 2.1. Antioxidant, AR Inhibition, and HPLC Profile of the Extracts and Fractions of L. meyenii

Previously we found that the 70% MeOH extract of *L. meyenii* exhibited strong DPPH radical scavenging activity and AR inhibitory activity [15]. In order to discover the active components in *L. meyenii*, the activities of the CH_2_Cl_2_, MeOH, and 50% MeOH extracts of *L. meyenii* against DPPH radicals and AR were comparatively determined using quercetin as a positive control [23,24]. As shown in Table 1, both the antioxidant and AR inhibitory activities of the MeOH extract (DPPH, IC_50_ 32.81 µg/mL; AR, IC_50_ 1.64 µg/mL) and the 50% MeOH extract (DPPH, IC_50_ 34.04 µg/mL; AR, IC_50_ 4.02 µg/mL) were significantly higher than those of the CH_2_Cl_2_ extract (DPPH, 16.54% inhibition at 40 µg/mL; AR, 11.8% inhibition at 10 µg/mL). Moreover, the MeOH extract and the 50% MeOH extract showed higher AR inhibitory activity than quercetin (IC_50_ 4.34 µg/mL) and lower DPPH scavenging activity than quercetin (IC_50_ 10.46 µg/mL). The HPLC profile of the extracts suggested that the antioxidant and AR inhibitory activities of the MeOH extract and the 50% MeOH extract were mainly due to one major compound (Figure 1). 

Then, the MeOH extract and the 50% MeOH extract were further partitioned using H_2_Cl_2_, EtOAc, *n*-BuOH, and water and subjected to activity assay (Table 1). Among these fractions, the EtOAc fraction of the 50% MeOH extract showed the highest DPPH scavenging activity (IC_50_ 14.81 µg/mL) and AR inhibitory activity (IC_50_, 0.86 µg/mL), followed by the EtOAc fraction of the MeOH extract (DPPH, IC_50_ 15.48 µg/mL; AR, 1.24 µg/mL), the *n*-BuOH fraction of the 50% MeOH (DPPH, IC_50_ 26.20 µg/mL; AR, 1.23 µg/mL), and the *n*-BuOH fraction of the MeOH extract (DPPH, IC_50_ 31.54 µg/mL; AR, 1.94 µg/mL), all of which exhibited significantly higher AR inhibitory activity than quercetin (IC_50_ 4.34 µg/mL), indicating that strong AR inhibitors exist in these fractions. Moreover, potent antioxidants may also exist in these fractions despite their DPPH scavenging activities being lower than those of quercetin (IC_50_ 10.46 µg/mL). Further HPLC profiles of the fractions and the white-color precipitate produced during the partition process of the MeOH extract revealed more components in addition to the major one (Figure 2). To better understand the component composition and discover highly active antioxidants and AR inhibitors, the minor compounds **1**, **2**, **4**–**7**, together with the major compound **3**, were selected as the target compounds to be separated (Figure 2).

### 2.2. Separation of the Phytochemicals in the Active Fractions of L. meyenii

#### 2.2.1. Separation of Components from the EtOAc Fraction of 50% MeOH Extract by HSCCC

Selection of a suitable solvent system plays a pivotal role to achieve a successful HSCCC separation and an ideal solvent system usually offers a partition coefficient (*K*) within 0.5 and 2.0 (0.5 ≤ *K* ≤ 2.0) and a separation factor (*α*) more than 1.5 (*α* ≥ 1.5; *α* = *K*_1_/*K*_2_, *K*_1_ ≥ *K*_2_) [22]. Accordingly, suitable *K* values were obtained from *n*-hexane/EtOAc/MeOH/water (2:5:2:5, *v*/*v*) for compounds **1** and **3** (*K*_1_ = 1.16, *K*_3_ = 0.90); from *n*-hexane/EtOAc/MeOH/water (3:5:3:5, *v*/*v*) for compound **5** (*K* = 0.67); and from *n*-hexane/EtOAc/MeOH/water (4:5:4:5, *v*/*v*) for compound **7** (*K* = 0.68) (Table 2). However, compounds **1** and **3** exhibited a small α value (*α_K_*_1/*K*3_ = 1.29) using *n*-hexane/EtOAc/MeOH/water (2:5:2:5, *v*/*v*). Subsequently, modification of the solvent system by adding acid, a widely used strategy for HSCCC separation [22,25], was performed by adding acetic acid (0.1%, *v*/*v*) to the solvent system *n*-hexane/EtOAc/MeOH/water (2:5:2:5, *v*/*v*). However, this did not improve the *α* value of compounds **1** and **3** (*α_K_*_1/*K*3_ = 1.27). 

In addition to modification by adding acid, modification of solvent systems by adding MeOH has been recently proved to be a promising strategy for HSCCC separation [26,27], and was thus applied to modify the solvent system *n*-hexane/EtOAc/MeOH/water (2:5:2:5, *v*/*v*) in this study. Briefly, PL, the lower layer of *n*-hexane/EtOAc/MeOH/water (2:5:2:5, *v*/*v*), was modified by adding extra volume of 10%, 20%, and 40% MeOH. The MeOH-modified PLs were then individually paired with PU, the upper layer of *n*-hexane/EtOAc/MeOH/water (2:5:2:5, *v*/*v*), to form new solvent systems. As shown in Table 3, suitable *K* values of compounds **1**, **3** and **5** were achieved (*K*_1_ = 0.85, *K*_3_ = 0.58, *K*_5_ = 1.42) by the new solvent system paired by equal volumes of PU and PL + 10% MeOH (*v*/*v*), and the α value between compounds **1** and **3** increased to 1.47 (*K*_1_/*K*_3_). With 40% MeOH added to PL (*v*/*v*), resulting in the new solvent system, paired with an equal volume of PU and PL + 40% MeOH (*v*/*v*), the *K* value of compound **4** decreased to 0.67. A polarity-gradient elution HSCCC separation strategy was thus proposed by using PU as the stationary phase, whereas PL + 10% MeOH (*v*/*v*) was selected as the first mobile phase to separate components **1**, **3**, and **5**, and PL + 40% MeOH (*v*/*v*) was selected as the second mobile phase to separate component **7**.

Separation using a polarity-gradient elution HSCCC strategy was carried out as described in Section 4.3. Briefly, the EtOAc fraction of the 50% MeOH extract (1.37 g) was first eluted using PL + 10% MeOH (*v*/*v*) (the first mobile phase), isolating a single compound **5** (31.3 mg) and a mixture of components **1**, **3**, and **5** (846.1 mg) and further eluted using PL + 40% MeOH (*v*/*v*) (the second mobile phase) yielding compound **7** (33.0 mg) (Figure 3A,B). A severe loss of the stationary phase, a common problem in polarity-gradient elution HSCCC separation, as mentioned by [28], also occurred in this study, resulting in a retention rate of the stationary phase of only 20%. The purities of compounds **5** and **7**, determined by HPLC at 254 nm, were 89% and 95%, respectively. However, it failed to separate compounds **1** and **3** despite their α value being acceptable (*α_K_*_1_/*_K_*_2_ = 1.47). The reason for the failure to separate compounds **1** and **3** may be the poor stationary phase volume retention ratio (20%) [22] and overloading of the sample (1.37 g), particularly the major compound **3**, because the elution of the minor compound **1** after that of the major compound **3** may be overlapped by the “tail” of the major compound **3**. The mixture of compounds **1**, **3**, and **5** (806.9 mg) was subjected to a second run of HSCCC using PL + 10% MeOH (*v*/*v*) as the stationary phase and PU as the mobile phase, as described in Section 4.3, completely separating compounds **1** (24.1 mg), **3** (607.1 mg), and **5** (21.3 mg) with purities of 97%, 99%, and 96% by HPLC detection at 254 nm (Figure 3C,D). Moreover, the retention rate of the stationary phase increased to approximately 60%, which contributed to a better separation resolution (Figure 3C).

#### 2.2.2. Pre-HPLC Separation of the Components in the H_2_Cl_2_ Fraction of 50% MeOH Extract and the Components in the Partition Precipitate of MeOH Extract

With a simple component composition, the H_2_Cl_2_ fraction of the 50% MeOH extract was used to separate compound **6** using pre-HPLC, as described in Section 4.4. From 100 mg of the H_2_Cl_2_ fraction, 20.6 mg of compound **3** and 16 mg of compound **6** were separated (Figure 4A,B). The partition precipitate of the MeOH extract was used to separate compounds **2** and **4** due to relatively high content and simple component composition. Because the partition precipitate showed very low solubility in all the HSCCC solvent systems tested, it was separated by pre-HPLC, as described in Section 4.5, resulting in separation of 8.8 mg of compound **2** and 6.9 mg of compound **4** from 37.6 mg of the partition precipitate (Figure 4C,D). Finally, compounds **1**–**7** were all separated.

The extraction, partition, and separation procedures are summarized in Figure 5 to present a clear experimental process.

### 2.3. Structure Identification of the Separated Compounds ***1***–***7***

The structures of the separated compounds from *L. meyenii* (**1**–**7**) are shown in Figure 6. Compounds **1**–**7** were identified via NMR (Appendix A), EI-MS and ESI-MS/MS analysis, and by comparison with previously published papers. The MS information of all the compounds is listed as follows. 

Caffeic acid (**1**): yellow powder; EI-MS fragments (*m*/*z*) and intensity (%): 180 (100.00%), 163 (35.03%), 136 (86.47%), and 89 (65.64%). The ^1^H NMR (400 MHz, MeOD-*d*_4_), as summarized in Appendix A, was identical to [29]. The raw ^1^H NMR spectrum is listed in Appendix A.

Hesperidin (**2**): white powder; ESI-MS/MS *m*/*z*: negative ion, primary mass spectrum (MS) ion [M-H]^−^ 609.1; major fragment ions of the secondary mass spectrum (MS/MS) from [M-H]^−^, 609.1, 343.3, 325.3, 301.1, and 286.0 [30]. Positive ion, primary MS ion [M+H]^+^ 611.5; major fragment ions of MS/MS from [M-H]^+^ 611.5 and 303.2 [31]. The ^1^H NMR (600 MHz, DMSO-*d*_6_), as summarized in Appendix A, was identical to [32,33]. The raw ^1^H NMR, ^1^H-^1^H COSY NMR spectra, and ESI-MS/MS data are listed in Appendix A.

Rosmarinic acid (**3**): yellowish powder; EI-MS fragments (*m*/*z*) and intensity (%): 360 (0.39%), 212 (11.94%), 198 (17.14%), 194 (15.64%), 180 (37.96%), 179 (17.17%), 163 (19.33%), 136 (100%), 123 (98.35%), 107 (14.15%), 89 (27.00%), 77 (26.93%), 51 (10.64). The ^1^H NMR (600 MHz, MeOH-*d*_4_), as summarized in Appendix A, was identical to [34]. The raw ^1^H NMR and ^1^H-^1^H COSY NMR spectra are listed in Appendix A.

Diosmin (**4**): white powder; ESI-MS/MS *m*/*z*: negative ion, primary mass spectrum (MS) ion [M-H]^−^ 607.2; major fragment ions of the secondary mass spectrum (MS/MS) from [M-H]^−^, 607.2, 299.2, 284.2, 255.0, 227.0, and 151.0. Positive ion, primary MS ion [M+H]^+^ 301.2; major fragment ions of MS/MS from [M-H]^+^ 609.2, 463.2, 301.2, 286.2, 258.2, 229.2, and 153.0 [35]. The ^1^H NMR (600 MHz, DMSO-*d*_6_), as summarized in Appendix A, was identical to [33]. The raw ^1^H NMR, ^1^H-^1^H COSY NMR spectra, and ESI-MS/MS data are listed in Appendix A.

Methyl rosmarinate (**5**): yellowish powder; EI-MS fragments (*m*/*z*) and intensity (%): 374 (1.31%), 279 (16.73%), 167 (31.90%), 149 (100.00%), 127 (10.66%), 113 (20.98%), 112 (15.08%), 97 (18.87%), 85 (20.68), 71 (38.05), 57 (51.97). The ^1^H NMR (600 MHz, MeOH-*d*_4_), as summarized in Appendix A, was identical to [34,36]. The raw ^1^H NMR spectrum is listed in Appendix A.

Diosmetin (**6**): yellow powder; ESI-MS/MS *m*/*z*: negative ion, primary mass spectrum (MS) ion [M-H]^−^ 299.2; major fragment ions of the secondary mass spectrum (MS/MS) from [M-H]^−^, 299.2, 284.2, 256.0, 227.0, 151.0, 133.0, and 107.0. Positive ion, primary MS ion [M+H]^+^ 301.2; major fragment ions of MS/MS from [M-H]^+^ 301.2, 286.2, 258.2, 258.2, 229.2, 203.0, and 153.0 [35]. The ^1^H NMR (600 MHz, DMSO-*d*_6_) and the ^13^C NMR (151 MHz, DMSO-*d*_6_) as summarized in Appendix A, were identical to [37]. The raw ^1^H NMR, ^13^C NMR spectra, and ESI-MS/MS data are listed in Appendix A.

*n*-Butyl rosmarinate (**7**): yellowish powder; EI-MS fragments (*m*/*z*) and intensity (%): 416 (0.14%), 302 (14.80%), 254 (100.00%), 236 (59.02%), 180 (62.07%), 163 (21.30%), 153 (43.69%), 135 (25.10%), 123 (96%), 107 (30.75%), 77 (39.12%), 57 (21.53%, 51 (10.08). The ^1^H NMR (400 MHz, MeOH-*d*_4_), as summarized in Appendix A, was identical to [34]. The raw ^1^H NMR spectrum is listed in Appendix A.

### 2.4. Synthesis, Purification, and Structural Identification of Rosmarinic Acid Ethyl and Propyl Esters

To study the esterification effects of rosmarinic acid on its antioxidant and AR inhibitory activities, ethyl and propyl rosmarinates were further synthesized and separated, as described in Section 4.6. As monitored by HPLC, the esterification of rosmarinic acid with ethanol and propyl was almost completed within 72 h (Appendix A). A total of 73 mg of ethyl rosmarinate and 80 mg of propyl rosmarinate were obtained after the reaction solutions were centrifuged, filtered, and evaporated. However, the resulting compounds were not pure (Appendix A) and were, therefore, purified by HSCCC, as described in Section 4.6, yielding high-purity ethyl rosmarinate (41.7 mg; Appendix A) and propyl rosmarinate (37.2 mg; Appendix A). The synthetic structures are listed in Figure 6 and confirmed as follows.

Ethyl rosmarinate (synthetic compound **1**, **S1**): yellowish powder; EI-MS fragments (*m*/*z*) and intensity (%): 388 (8.04), 226 (82.82), 209 (100.00), 180 (97.46), 163 (93.90), 153 (37.63), 135 (59.98%), 123 (64.24), 107 (25.03), 89 (36.62), 77 (52.79), 51 (15.14). The ^1^H NMR (400 MHz, MeOH-*d*_4_), as summarized in Appendix A, was identical to [34]. The raw ^1^H NMR spectrum is listed in Appendix A.

Propyl rosmarinate (synthetic compound **2**, **S2**): yellowish powder; EI-MS fragments (m/z) and intensity (%): 402 (3.27), 240 (100.00), 222 (95.39), 180 (97.63), 163 (97.65), 153 (67.18), 135 (38.98), 124 (82.11), 107 (31.60), 89 (16.86), 77 (51.68), 51 (13.74). The ^1^H NMR (400 MHz, MeOH-*d*_4_), as summarized in Appendix A, was identical to [34]. The raw ^1^H NMR spectrum is listed in Appendix A.

### 2.5. Antioxidant and AR Inhibitory Activities of the Separated and Synthesized Compounds

Overall, rosmarinic acid and its methyl to *n*-butyl esters exhibited remarkable antioxidant activity (DPPH, IC_50_ 30.02–36.91 µM) and AR inhibitory activity (IC_50_ 1.02–4.08 µM), which were higher than or similar to those of quercetin (DPPH, IC_50_ 33.19 µM; AR, IC_50_ 16.16 µM); whereas caffeic acid, hesperidine, diosmin, and diosmetin were less active against DPPH radicals (2.61–43.33% inhibition at 50 µM) and AR (11.63–33.95% inhibition at 50 µM) (Table 4). In particular, the antioxidant and AR inhibitory activities of rosmarinic acid improved significantly after natural or synthetic esterification with MeOH, EtOH, 1-propanol, and *n*-BuOH. The antioxidant activities followed the order *n*-butyl rosmarinate > ethyl rosmarinate > propyl rosmarinate > methyl rosmarinate > rosmarinic acid, and their AR inhibitory activities followed the order ethyl rosmarinate > methyl rosmarinate > propyl rosmarinate > *n*-butyl rosmarinate > rosmarinic acid. However, the DPPH scavenging activities between ethyl and *n*-butyl rosmarinates and the AR inhibitory activities among methyl, ethyl, propyl, and *n*-butyl rosmarinates were not significantly different (*p* > 0.05). 

### 2.6. Quantification of Rosmarinic Acid in the Extracts and Fractions of L. meyenii

Possessing remarkable antioxidant and AR inhibitory activities, the major compound rosmarinic acid (**3**) in the extracts and fractions of *L. meyenii* was further quantified to better understand the proportion of rosmarinic acid and its contribution to the antioxidant and AR inhibitory activities. The HPLC quantification method was first validated by assessing the linearity, limit of detection, limit of quantification, precision, and accuracy (spike test), as described in Section 4.10. The standard curve of rosmarinic acid showed good linearity (r^2^ = 1.00) within the concentrations determined (0.39–400 µg/mL; HPLC injection volume 10 µL), and the limit of detection and limit of quantification of rosmarinic acid were 0.15 and 0.39 µg/mL, respectively (Appendix A). The precision accessed by relative standard deviation was between 1.24% (100.00 µg/mL) and 3.53% (12.50 µg/mL) in the intra-day test, and between 1.47% (100.0 µg/mL) and 4.85% (12.50 µg/mL) in the inter-day test. The accuracy assessed using the spike recovery test was between 103.17% and 105.46% (Appendix A). The quantification method was therefore validated by the obtained results and then applied to quantify rosmarinic acid. 

As listed in Table 5, rosmarinic acid was rich in the MeOH extract (33.84 mg/g) and the 50% MeOH extract (135.41 mg/g) of *L. meyenii*. In contrast, only a small amount of rosmarinic acid was present in the H_2_Cl_2_ extract (1.24 mg/g). Notably, after the partition of the MeOH extract, the rosmarinic acid content further increased to 804.07 mg/g and 373.71 mg/g in the EtOAc and BuOH fractions of the MeOH extract, respectively. Similarly, after partitioning the 50% MeOH extract, the contents of rosmarinic acid also increased in the EtOAc fraction (634.22 mg/g) and *n*-BuOH fraction (426.22 mg/g) of the 50% MeOH extract. The contents of rosmarinic acid in the other fractions of the MeOH and 50% MeOH extracts are listed in Table 5. Additionally, the content of rosmarinic acid in the dried raw material (aerial parts) of *L. meyenii* was calculated to be 37.22 mg/g (Table 5).

## 3. Discussion

We previously reported that *L. meyenii* showed strong antioxidant and AR inhibitory activities [15]. In this study, we proved that rosmarinic acid (**3**), methyl rosmarinate (**5**), and butyl rosmarinate (**7**) are the main active compounds in *L. meyenii* with remarkable antioxidant and AR inhibitory activities. In particular, rosmarinic acid is the key contributor to the antioxidant and AR inhibitory activities of *L. meyenii*, which is rich in the MeOH extract (333.84 mg/g) and 50% MeOH extract (135.41 mg/g) of *L. meyenii*, and is especially abundant in the EtOAc and *n*-BuOH fractions (373.71–804.07 mg/g) of the MeOH and 50% MeOH extracts. 

Herbal medicine plays an important role in the treatment of diabetes. Yet the underlying bioactive compounds of some plants are still unclear. In this study, we provided the HPLC profiles of all the extracts (Figure 1) and fractions (Figure 2) of *L. meyenii*, identified seven main compounds in its active extracts and fractions (Figure 6), and quantified the contents of the principal compound, rosmarinic acid (**3**), in all the extracts and fractions of *L. meyenii* (Table 5). These findings provide scientific evidence confirming its phytochemical composition and promotes its application. Notably, rosmarinic acid (**3**) is rich in the MeOH and 50% MeOH extracts (135.41–333.84 mg/mL), and the contents of rosmarinic acid in the EtOAc and BuOH fractions of the MeOH and 50% MeOH extracts remarkably increased (373.71–804.07 mg/g) after the simple solvent-solvent fractionation process (Table 5). Moreover, the content of rosmarinic acid in the dried raw material of *L. meyenii* was calculated to be 37.22 mg/g (Table 5), which is comparable to many popular rosmarinic acid-rich plants including *Salvia officinalis* (8.5–14.1 mg/g), *Rosmarinus officinalis* (10–11 mg/g), *Mentha spicata* L. (7.1–14.3 mg/g), and *Melissa officinalis* L. (27.4 mg/g) [38]. This indicates that *L. meyenii* is a promising source for the industrial production of rosmarinic acid. Rosmarinic acid (**3**) was identified as the key contributor to the antioxidant and AR inhibitory activities of *L. meyenii*, as the rosmarinic acid content (mg/g) showed a strong positive and significant correlation with the activities against DPPH radicals (*r* = 0.945, *p* < 0.001) and AR (*r* = 0.923, *p* < 0.001) in the extracts/fraction of *L. meyenii* (Figure 7). In addition to rosmarinic acid (**3**), six more compounds were separated and identified from *L. meyenii*, including caffeic acid (**1**), hesperidin (**2**), diosmin (**4**), methyl rosmarinate (**5**), diosmetin (**6**), and butyl rosmarinate (**7**), among which caffeic acid, rosmarinic acid, and methyl rosmarinate have previously been reported in *L. meyenii* [16,39]. In contrast, hesperidin, diosmin, diosmetin, and butyl rosmarinate were identified in *L. meyenii* for the first time in this study. Among these identified compounds, rosmarinic acid derivatives (**3**, **5**, **7**) showed higher antioxidant and AR inhibitory activities than the other compounds (**1**, **2**, **4**, **6**) (Table 4). Nevertheless, hesperidin (**2**) [40,41], diosmin (**4**) [42,43], and diosmetin (**6**) [44] also show anti-diabetic properties with diabetic neuroprotective and antihyperglycemic effects or via up-regulating the IRS/PI3K/AKT signaling pathway. Our study adds scientific evidence to the existing literature about this traditional anti-diabetic herbal medicine, *L. meyenii* [8], its antioxidant properties, AR inhibitory activity, phytochemical composition, and rosmarinic acid content. 

Moreover, the present study demonstrated that esterification of rosmarinic acid with short-chain primary alcohols (C_1_–C_4_) significantly enhanced its antioxidant and AR inhibitory activities (Table 4). A previous study reported that esterification of rosmarinic acid using short to medium chain primary alcohols (C_4_, C_10_, and C_16_) can increase cell uptake and boost antioxidant activity without significant cytotoxicity [45], whereas a more recent study showed that only short-chain (≤C4) esterification of rosmarinic acid can increase its bioavailability, and esterification with longer alkyl chains leads to severe cytotoxicity [46]. However, no studies have been carried out to compare the antioxidant activity among rosmarinic acid and its short-chain esters (≤C4). Considering that rosmarinic acid, methyl rosmarinate, and butyl rosmarinate were previously separated from *L. meyenii* (Figure 3), we then synthesized ethyl and propyl rosmarinates (Appendix A). Furthermore, DPPH scavenging assay revealed that the antioxidant potential of rosmarinic acid (DPPH IC_50_ 36.91 µM) was significantly increased after being esterified using short-chain primary alcohols (C_1_–C_4_, DPPH IC_50_ 30.02–33.01 µM) (Table 4), among which ethyl rosmarinate (DPPH IC_50_ 30.54 µM) and butyl rosmarinate (DPPH IC_50_ 30.02 µM) exhibited the highest antioxidant potential (Table 4). Moreover, the AR inhibitory activity of rosmarinic acid (IC_50_ 4.08 µM) was also significantly increased after being esterified to methyl–butyl rosmarinates (IC_50_ 1.02–1.54 µM) (Table 4). Notably, ethyl rosmarinate (IC_50_ 1.02 µM) was four times more potent than rosmarinic acid (IC_50_ 4.08 µM) and approximately 16 times more potent than the positive control quercetin (IC_50_ 16.16 µM) [23]. Apparently, with ethyl rosmarinate as the node, increasing the chain length of the primary alcohols tends to reduce the AR inhibitory activity of rosmarinic acid esters despite there being no significant differences among methyl–butyl rosmarinates regarding AR inhibition (Table 4). In addition to esterification of rosmarinic acid by short-chain primary alcohols (≤C4), amination of rosmarinic acid using phenylmethanamine, 4-(aminomethyl)phenol, and 1-phenylethan-1-amine was also reported to improve its AR inhibitory activity [47], indicating that derivatization of the hydroxy group connected to 9′-C in rosmarinic acid is a promising strategy to improve its bioactivity regarding antioxidation, AR inhibition, and even protein kinase B (Akt) inhibition [48]. 

In addition, the HSCCC separation method established in this study is a promising method for preparative separation of rosmarinic acid from *L. meyenii* and other natural products. Chen et al. previously separated 1.9 mg of rosmarinic acid from *Salvia miltiorrhiza* Bunge (80 mg) by HSCCC using solvent system *n*-hexane/EtOAc/MeOH/water (1.5:5:5:1.5, *v*/*v*) [49]. Xie et al. selected *n*-hexane/EtOAc/MeOH/water (1:4:1:4, *v*/*v*) as the solvent system and separated 11 mg of rosmarinic acid from 100 mg of EtOAc extract of *Glechoma hederacea* L. by HSCCC [50]. Kwon et al. succeeded in separation of 20.4 mg of rosmarinic acid from 200 mg of EtOAc fraction of *Perilla frutescens* using step-wise HSCCC [51]. More recently, Zhu et al. separated 8 mg of rosmarinic acid from 160 mg of *L. meyenii* by HSCCC and pre-HPLC using *n*-hexane/EtOAc/MeOH/water (3:5:3:5 + 1.5% acetic acid, *v*/*v*) as the HSCCC solvent system [52]. Using PL + 10% MeOH (*v*/*v*) as the stationary phase and PU as the mobile phase, 607.1 mg of high-purity rosmarinic acid (99%) was separated from 806.9 mg of *L. meyenii* subfraction (Figure 3) in this study, where PL and PU are the abbreviations of the partitioned lower layer and partitioned upper layer of the solvent system *n*-hexane/EtOAc/MeOH/water (2:5:2:5, *v*/*v*), respectively.

## 4. Materials and Methods

### 4.1. Reagents and Plant

2,2-Diphenyl-1-picrylhydrazyl (DPPH), quercetin, DL-glyceraldehyde (dimer), *β*-nicotinamide adenine dinucleotide 2′-phosphate reduced tetrasodium salt hydrate (NADPH), glacial acetic acid, sodium phosphate dibasic dodecahydrate, trifluoroacetic acid, potassium phosphate monobasic, ethanol (99.8%), 1-propanol (99.7%), rosmarinic acid (97%), and sodium phosphate dibasic dehydrate were procured from Sigma-Aldrich Chemical Co. (St. Louis, MI, USA). An Amberlite^®^ IR-120 (H^+^ form) ion exchanger and ammonium sulfate were purchased from Merck KGaA Co. (Darmstadt, Germany). A 3A molecular sieve was purchased from Consolidated Chemical & Solvents LLC. (Quakertown, PA, USA). The other organic solvents were purchased from J. T. Baker Co. (Phillipsburg, NJ, USA), including HPLC grade for HPLC and preparative HPLC (Pre-HPLC) assays and analytical grade for extraction, fractionation, and HSCCC separations. The ultrapure water used in this study was produced using a Milli-Q water purification system (Millipore Co., Bedford, MA, USA).

The aerial parts of *L. meyenii* were collected from Lima, and the specimen was authenticated by Paul H. Gonzales Arce (P.H.G.A.). The dried material was placed at the Center for Efficacy Assessment and Development of Functional Foods and Drugs, Hallym University.

### 4.2. Extraction and Partition of L. meyenii

The dried aerial parts of *L. meyenii* (455 g) were successively extracted by 5 L of dichloromethane (CH_2_Cl_2_) (2 × 2 d), MeOH (2 × 2 d), and 50% MeOH aqueous solution (2 × 2 d) at room temperature (approximately 22–28 °C). The two extraction solutions of each extraction solvent were combined, filtered, and evaporated to dryness by rotary evaporation (37 °C), yielding 30.06 g of CH_2_Cl_2_ extract, 33.27 g of MeOH extract, and 42.78 g of 50% MeOH extract.

Then, the MeOH extract (3.12 g) was suspended in water (100 mL) assisted by sonication and partitioned twice by an equal volume of CH_2_Cl_2_, EtOAc, and *n*-BuOH to yield sub-fractions of CH_2_Cl_2_ (0.24 g), EtOAc (0.38 g), *n*-BuOH (1.11 g), and water (1.38 g). Notably, a white-color precipitate was produced during the partition process, which was separately collected and evaporated, yielding 37.6 mg of powder.

Similarly, the 50% MeOH extract was also subjected to partition procedure by suspending 30.16 g of extract in 1 L of water and partitioning the solution using CH_2_Cl_2_ (2 × 1 L), EtOAc (2 × 1 L), and *n*-BuOH (2 × 1 L) to yield sub-fractions of CH_2_Cl_2_ (0.46 g), EtOAc (2.73 g), *n*-BuOH (2.63 g), and water (21.97 g). 

### 4.3. Separation of Components from the EtOAc Fraction of 50% MeOH Extract by HSCCC

#### 4.3.1. Screening and Modification of HSCCC Solvent System

As described previously, screening of HSCCC solvent systems, composed of *n*-hexane, EtOAc, MeOH, and water, was carried out [53]. Briefly, each solvent system was prepared, thoroughly mixed, and divided into upper and lower phases after settling. Then a proper amount of sample (the EtOAc fraction of the 50% MeOH extract; 0.1–0.5 mg) was weighed in a 1.5 mL tube and dissolved by 1 mL of a solvent composed of 500 µL of upper phase and 500 µL of lower phase. The sample solution was thoroughly mixed by a vortex to equilibrate the contents. After settling, equal volumes of the upper and lower layers (each 200 µL) of the sample solution were transferred respectively to new 1.5 mL tubes and evaporated by nitrogen gas, which were then re-dissolved using 200 µL of MeOH and subjected to HPLC detection (injection volume 10 µL). The *K* value is calculated as *A*_upper_/*A*_lower_, where *A*_upper_ and *A*_lower_ are the HPLC peak areas of a compound in the upper and lower layers, respectively. However, the solvent systems tested could not provide satisfactory *K* values and *α* values, which were further modified as follows.

Adding 0.1% volume of acetic acid to *n*-hexane/EtOAc/MeOH/water (2:5:2:5, *v*/*v*) was first carried out to modify, but failed to improve, the solvent system, which was further modified by adding MeOH. Briefly, the solvent system *n*-hexane/EtOAc/MeOH/water (2:5:2:5, *v*/*v*) was prepared and partitioned into the upper layer (PU) and lower layer (PL). Next, a 10–40% volume of MeOH was added to PL to obtain MeOH-modified PLs. As mentioned above, the MeOH-modified PLs were individually paired with PU to form new solvent systems for determining *K* values and α values. Finally, PL + 10% MeOH (*v*/*v*) and PL + 40% MeOH (*v*/*v*) were selected to pair new solvent systems with PU to separate the compounds from the EtOAc fraction of the 50% MeOH extract in polarity-gradient and polarity-constant manners.

#### 4.3.2. Preparation of Solvent System and HSCCC Separation

The solvent system *n*-hexane/EtOAc/MeOH/water (2:5:2:5, *v*/*v*) was prepared and separated into PL and PU using a funnel. Then PL + 10% MeOH (*v*/*v*) and PL + 40% MeOH (*v*/*v*) were prepared by adding extra MeOH to PL. Solvents PU, PL + 10% MeOH (*v*/*v*), and PL + 40% MeOH (*v*/*v*) were degassed by sonication for 30 min before use. A TBE 300C HSCCC (Tauto Biotech. Co., Ltd., Shanghai, China) with three preparative coils (diameter 2.6 mm; total volume 300 mL) was used for separation. An Isolera FLASH purification system (Biotage, Uppsala, Sweden) was fitted to the HSCCC machine as a pump, UV monitor, and fraction collector.

HSCCC separation of the sample was first performed in a polarity-gradient elution manner. Briefly, PU was used as the stationary phase to completely fill the HSCCC coil, and the rotational speed was adjusted to 900 rpm. Subsequently, solvent PL + 10% MeOH (*v*/*v*) was introduced as the first mobile phase at 3 mL/min until a hydrodynamic equilibrium was achieved. Then, the sample solution (15 mL) was loaded to the sample loop (maximum 20 mL), which was prepared by dissolving 1.37 g of the EtOAc fraction of the 50% MeOH extract in 15 mL of biphasic solvents composed of 7 mL of PU and 8 mL of PL + 10% MeOH (*v*/*v*). Next, the sample was eluted (3 mL/min) by PL + 10% MeOH (*v*/*v*) (the first mobile phase; 0–360 mL) for compounds **1**, **3**, and **5**, and eluted (3 mL/min) by PL + 40% MeOH (*v*/*v*) (the second mobile phase; 360–510 mL) for compound **7**. The eluate was monitored at 254 nm. After completing the separation, the solvent was pumped out in air and collected by a graduated cylinder to calculate the retention ratio of the stationary phase, which was calculated as *V*s/*V*c, where *V*s is the stationary phase volume retained in the column coil, and *V*c is the HSCCC column coil volume (300 mL).

Components **1**, **3**, and a small amount of **5** were concentrated as a mixture by the polarity-gradient elution HSCCC, which were further separated by polarity-constant elution HSCCC using PL + 10% MeOH (*v*/*v*) as the stationary phase and PU as the mobile phase. Briefly, the HSCCC coil was filled with solvent PL + 10% MeOH (*v*/*v*), and the rotational speed was adjusted to 900 rpm. Next, PU was pumped in at 3 mL/min until a hydrodynamic equilibrium was achieved. Subsequently, 0.81 g of the mixture fraction, mainly composed of **1**, **3**, and **5**, was dissolved in 15 mL of biphasic solvents consisting of 7 mL of PU and 8 mL of PL + 10% MeOH (*v*/*v*) and loaded for HSCCC separation. The mobile phase elution speed was 3 mL/min, and the eluate was monitored at 254 nm.

### 4.4. Separation of the Components in the H_2_Cl_2_ Fraction of 50% MeOH Extract by Pre-HPLC

The separation was performed on a pre-column (Φ20 × 500 mm; JAIGEL-GS310) equipped with LC-908 preparative HPLC (JAI, Japan). Approximately 100 mg of the H_2_Cl_2_ fraction of the 50% MeOH extract was dissolved in 1.5 mL of MeOH, filtered (0.22 µm; Millipore Millex-GP, Bedford, MA, USA), and loaded to a sample loop (maximum volume 2 mL) for purification, which was successively eluted by 50% MeOH (0–1000 mL; 4 mL/min) and 60% MeOH (1000–1600 mL; 4 mL/min), and monitored at 254 nm.

### 4.5. Separation of the Components in the Partition Precipitate of MeOH Extract by Pre-HPLC

The partition precipitate of the MeOH extract was separated using the same pre-HPLC and column as mentioned above. Briefly, 37.6 mg of the sample was dissolved in 800 µL of 75% DMSO aqueous solution and loaded for purification using 65% MeOH (0–450 mL) as the mobile phase and eluted at 4 mL/min. The elution was monitored at 254 nm.

### 4.6. Synthesis and Purification of Rosmarinic Acid Ethyl and Propyl Esters

The ethyl and propyl rosmarinates were synthesized as described previously [34]. The 3A molecular sieve was activated by heating at approximately 350 °C for 3.5 h in a muffle furnace, and the Amberlite^®^ IR-120 (H^+^ form) acidic sulfonic resin was activated by heating at 110 °C for 48 h. The activated 3A molecular sieve was individually mixed with ethanol and 1-propanol (10% *w*/*v*) and allowed to stand for 48 h for solvent dehydration. Then, the anhydrous ethanol and 1-propanol (each 20 mL) were individually mixed with the activated 3A molecular sieve (3 g), activated Amberlite^®^ IR-120 acidic sulfonic resin (1 g), and rosmarinic acid (100 mg) in sealed reagent bottles. The reaction mixtures were incubated at 55 °C in an orbital shaker (145 rpm), and the reaction solutions were continuously monitored using HPLC at 0, 14, 24, 37, and 72 h. The HPLC samples were prepared by mixing 10 µL of each reaction solution and 800 µL of MeOH, filtered (0.45 µm; Whatman, Clifton, NJ, USA), and subjected to HPLC detection (injection volume 15 µL). All reactions were completed within 72 h, and the reaction solutions were then centrifuged (Union 32R Plus centrifuge; Hanil Scientific Inc., Gimpo, Korea) for 30 min at 4000 rpm (3720× *g*) and 25 °C. After centrifugation, each supernatant was further filtered (0.45 µm syringe filter) and evaporated by rotary evaporation and a Genevac EZ-2 Plus evaporator (SP-Scientific, Gardiner, NY, USA), affording 73 mg of ethyl rosmarinate and 80 mg of propyl rosmarinate. 

Further purification of the synthetic ethyl and propyl esters by HSCCC was carried out using *n*-hexane/EtOAc/MeOH/water (4:5:4:5, *v*/*v*) as the solvent system, which offered suitable *K* values for ethyl rosmarinate (*K* = 0.49) and propyl rosmarinate (*K* = 0.87). In brief, the HSCCC coil was filled with the upper layer of the solvent system as the stationary phase, and the rotation speed was then adjusted to 850 rpm. The lower layer of the solvent system was then introduced as the mobile phase at 4 mL/min until a hydrodynamic equilibrium was achieved. Then, the sample solution (15 mL) was loaded to the sample loop, which was prepared by dissolving each sample (ethyl rosmarinate, 72 mg; propyl rosmarinate, 78 mg) in 15 mL of biphasic solvents composed of 7 mL of stationary phase and 8 mL of the mobile phase. The eluates were monitored at 210 and 280 nm for ethyl rosmarinate, and 280 nm for propyl rosmarinate. The elution speed of the mobile phase was 4 mL/min for both of the HSCCC separations.

### 4.7. HPLC Condition

A Dionex system (Dionex, Sunnyvale, CA, USA) was used for HPLC analysis, which consisted of a P850 pump, an ASI-100 automated sample injector, an STH585 column oven (maintained at 30 °C), and a UVD170S detector. Acid water (0.1% trifluoroacetic acid) (A) and MeOH (B) were used as the HPLC mobile phases. The *L. meyenii* samples were monitored at 254 nm and separated (0.7 mL/min) using an Eclipse XDB-C18 column (150 × 4.6 mm, 5 µm) as follows: 10–100% B at 0–20 min; 100% B at 20–24 min; 100–10% B at 26–30 min; and the synthetic samples were monitored at 210 and 280 nm and separated using a Synergi Hydro-RP 80A column (150 × 4.60 mm, 4 μm) as follows: 10–55% B at 0–5 min; 55–100% B at 5–20 min; 100–10 % B at 20–25 min; 10% B at 25–30 min.

### 4.8. Structure Identification

The structural identification of the isolated natural compounds or synthetic compounds was performed using EI-MS (JEOL JMS-700; JEOL Ltd., Tokyo, Japan), an AB Sciex QTrap^®^ 5500 mass spectrometer (Foster City, CA, USA), 400 MHz ^1^H-NMR (JNM-ECZ400S/L1; JEOL Ltd., Japan), 600 MHz NMR (Bruker Avance Neo 600 Ultra Shield^TM^; Bruker Biospin, Rheinstetten, Germany), and comparison with published papers and a standard compound (rosmarinic acid).

### 4.9. Activity Assay

#### 4.9.1. Antioxidant Assay

The antioxidant potential of the samples was evaluated using DPPH radical scavenging assay, as reported previously [12]. In brief, 180 μL of freshly prepared DPPH solution (0.32 mM in MeOH) was mixed with 20 μL of the sample (in 50% MeOH, extracts and fractions, 50–400 µg/mL; components, 125–500 µM) in a 96-well plate and incubated for 20 min in the dark at 25 °C. Then, the absorbance (570 nm) of the reaction solution was measured using an EL800 microplate reader (Bio-Tek Instruments, Winooski, VT, USA). Quercetin was used as a positive control. The DPPH radical scavenging activity (%) was calculated using Equation (1):(1)$$\%\;inhibition=\left(1-\frac{A_{sample}-A_{blank1}}{A_{control}-A_{blank2}}\right)\times 100\%$$
where *A_sample_* is the absorbance of DPPH solution with the sample, *A_blank1_* is the absorbance of the test sample without DPPH, *A_control_* is the absorbance of DPPH solution without sample, *A_blank2_* is the absorbance of MeOH, without DPPH or sample.

#### 4.9.2. AR Inhibition Assay

The preparation of rat lens AR and AR inhibition assay were conducted as we previously reported [11]. The eyes of 10-week Sprague–Dawley rats (250–280 g) were removed and kept at −70 °C before use. Then, the lenses were removed from the eyes using surgical scissors and tweezers, ground in a mortar (precooled at −70 °C) and extracted using 0.1 M phosphate-buffered saline (PBS) of pH 6.2 (approximately 0.5 mL of buffer per one rat lens). The extract solution was further centrifuged at 10,000× *g* for 30 min at 4 °C (Mega 17R, Hanil Science Industry, Gimpo, Korea), and the supernatant was collected and used as rat lens AR homogenate. 

For AR inhibition assay, 100 µL of 0.1 M PBS (pH 7.0), 20 µL of AR homogenate, 20 µL of NADPH (cofactor, 2.4 mM in 0.1 M PBS of pH 8.0), 20 µL of the sample (in a mixture of water and DMSO; extracts and fractions 62.5–100 µg/mL, compounds 3.9–500 µM), and 20 µL of ammonium sulfate solution (4 M in 0.1 M PBS of pH 7.0) were pipetted into a 96-well plate. Then, 20 µL of the substrate (in 0.1 M PBS of pH 7.0; 25 mM of DL-glyceraldehyde dimmer) was added, and the absorbance (340 nm) was measured for 6 min using an Epoch microplate spectrophotometer (BioTek Instruments, Winooski, VT, USA). Quercetin was used as the positive control (final concentration 6.25–25 µM) [23]. DMSO was used to prepare samples, but its ratio was kept within 0.5% (*v*/*v*) of the reaction system. The AR inhibition (%) by samples was calculated using Equation (2):(2)$$\%\;inhibition=\left(1-\frac{\left|Slope_{s}\right|-\left|Slope_{b}\right|}{\left|Slope_{c}\right|-\left|Slope_{b}\right|}\right)\times 100\%$$
where *Slope*_s_, *Slope_b_*, and *Slope_c_* are the slopes derived from the OD340 nm (abscissa) versus the reaction time (min; ordinate)—dotted lines of sample group (with enzyme and sample), blank group (without enzyme or sample), and control group (with enzyme but without sample). |*Slope*| is the absolute value of the slope.

### 4.10. Quantification of Rosmarinic Acid in the Extracts and Fractions of L. meyenii

The contents of rosmarinic acid (**3**) in the extracts and fractions of *L. meyenii* were quantified using HPLC assay (injection volume 10 µL) as the condition described in Section 4.7. A stock solution of rosmarinic acid was prepared in 50% MeOH at 2.00 mg/mL and then diluted to appropriate concentrations using 50% MeOH to make a calibration curve and validate the HPLC method. The calibration curve was plotted with the HPLC peak areas as the *y*-axis and the concentrations of rosmarinic acid as the *x*-axis (triplicate, 0.39–400 µg/mL). The limit of detection and limit of quantification were determined by signal-to-noise ratios of three (S/N = 3) and ten (S/N = 10), respectively. The precision of the quantification method was evaluated by measuring the relative standard deviation (RSD) values of the peak areas of rosmarinic acid (12.5 and 100 µg/mL) determined by HPLC at intra-day (*n* = 6) and inter-day (*n* = 3). To examine the accuracy of the quantification method, a spike recovery test was carried out by mixing 0.20 mL of rosmarinic acid standard solutions (in 50% MeOH, 25 and 200 µg/mL) individually with 0.20 mL of the 50% MeOH extract solution (in 50% MeOH, 100 µg/mL). The peak areas of rosmarinic acid in the 50% MeOH extract solution and the spiked solution were determined by HPLC assay (injection volume 10 µL, in triplicate) to calculate the concentrations of rosmarinic acid in the 50% MeOH extract solution (*C*_1_) and the spiked solution (*C*_2_) using the plotted calibration curve, which were used to calculate the spike recovery using Equation (3):(3)$$\%\;\mathrm{Spike}\;\mathrm{recovery}=\frac{C_{2}\times V_{2}-C_{1}\times V_{1}}{C_{0}\times V_{0}}\times 100\%$$
where *V*_0_ and *V*_1_ are the volumes of the standard solution (*V*_0_, 0.20 mL) and the 50% MeOH extract solution (*V*_1_, 0.20 mL) used for the spiking test, respectively, and *V*_2_ is the volume of the spiked sample solution (*V*_2_, *V*_2_ = *V*_0_ + *V*_1_ = 0.4 mL); *C*_0_, *C*_1_, and *C*_2_ are the concentrations of rosmarinic acid in the standard solution (*C*_0_, 25 and 200 µg/mL), the 50% MeOH extract solution (*C*_1_, calculated), and the spiked solution (*C*_2_, calculated). 

In addition to the 50% MeOH extract solution, the other extracts and fractions of *L. meyenii* were also prepared as solutions (in 50% MeOH, 100 µg/mL) and determined by HPLC (injection volume 10 µL, in triplicate). The content (mg/g) of rosmarinic acid in each sample powder was calculated as the concentration of the rosmarinic acid in the sample solution (calculated from the calibration curve) divided by the concentration of the sample solution tested (100 µg/mL). 

### 4.11. Statistical Analysis

All activity assays were performed in triplicate and the results were presented as mean ± standard deviations (SDs). The half-maximal inhibitory concentrations (IC_50_ values) of samples against DPPH radicals and AR were calculated via linear regression and logarithmic analysis, respectively. One-way ANOVA with Tukey’s multiple comparisons test was used to compare the differences of the IC_50_ values of the active samples against DPPH radicals and AR, which were performed using GraphPad Prism (Version 8.4.2, GraphPad Software, San Diego, CA, USA), and *p* < 0.05 was statistically significant. Moreover, the correlations of rosmarinic acid content (mg/g) and the activities of DPPH radical scavenging and AR inhibition in the extracts/fraction of *L. meyenii* were assessed by calculating Pearson’s correlation coefficients with SPSS software (Version 25; IBM, New York, NY, USA).

## 5. Conclusions

In conclusion, seven compounds were separated and identified from the MeOH and 50% MeOH extracts of *L. meyenii* and their active fractions, namely, caffeic acid (**1**), hesperidin (**2**), rosmarinic acid (**3**), diosmin (**4**), methyl rosmarinate (**5**), diosmetin (**6**), and butyl rosmarinate (**7**). Among these, compounds **2**, **4**, **6**, and **7** are reported in *L. meyenii* for the first time and **3**, **5**, and **7** possessed remarkable antioxidant and AR inhibitory activities. In particular, **3** is the key contributor to the antioxidant and AR inhibitory activities of *L. meyenii*, which was rich in the MeOH extract (333.84 mg/g) and 50% MeOH extract (135.41 mg/g) of *L. meyenii.* It was especially abundant in the EtOAc and *n*-BuOH fractions (373.71–804.07 mg/g) of the MeOH and 50% MeOH extracts. Moreover, comparative study of rosmarinic acid and its short-chain esters (≤C4) revealed that esterification of rosmarinic acid using short-chain primary alcohols (≤C4) can significantly increase its antioxidant and AR inhibitory potential. In addition, the HSCCC separation method established in this study can be used for preparative separation of rosmarinic acid from *L. meyenii* and other natural products.

## Figures and Tables

**Figure 1 plants-10-02773-f001:**
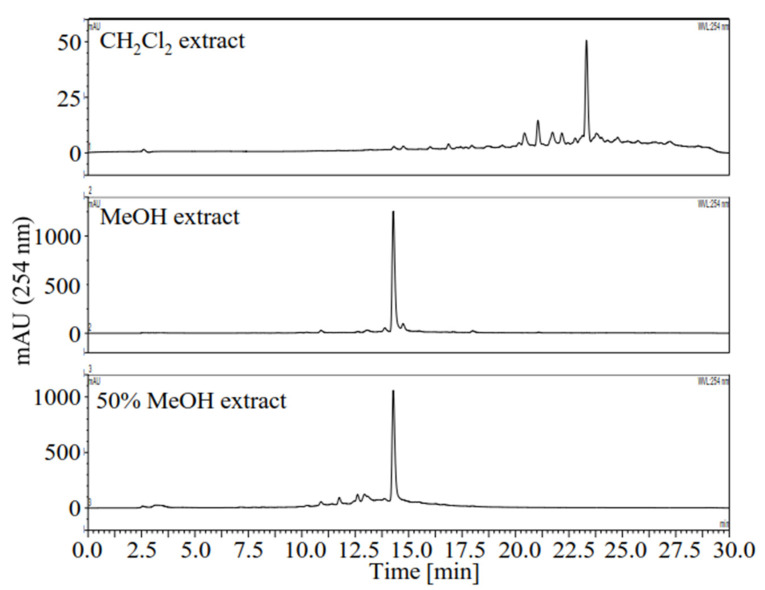
HPLC profiles of the extracts of *Lepechinia meyenii* (Walp.) Epling. The major compound in the MeOH and 50% MeOH extracts was later identified as rosmarinic acid.

**Figure 2 plants-10-02773-f002:**
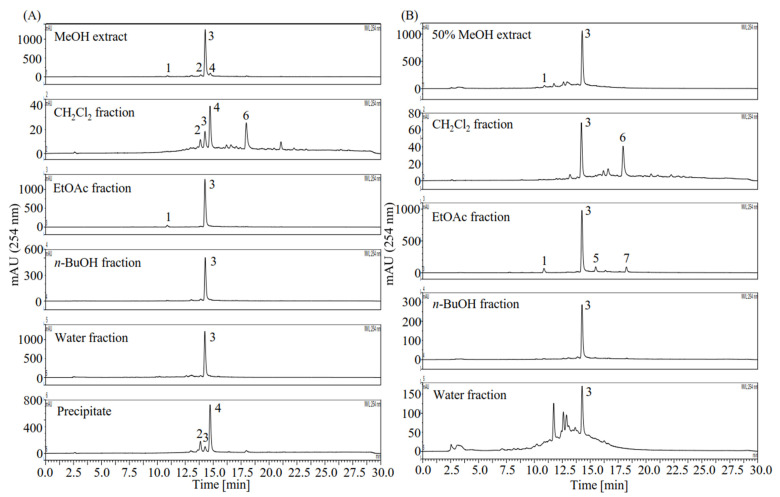
HPLC profiles of the fractions from the MeOH and the 50% MeOH extracts of *Lepechinia meyenii* (Walp.) Epling. (**A**) HPLC profiles of the MeOH extract and its partitioned fractions and precipitate. (**B**) HPLC profiles of the 50% MeOH extract and its partitioned fractions. The precipitate in (**A**) was produced during the partition process of the MeOH extract. Notably, compounds **1**–**7** were later identified as caffeic acid (**1**), hesperidin (**2**), rosmarinic acid (**3**), diosmin (**4**), methyl rosmarinate (**5**), diosmetin (**6**), and butyl rosmarinate (**7**).

**Figure 3 plants-10-02773-f003:**
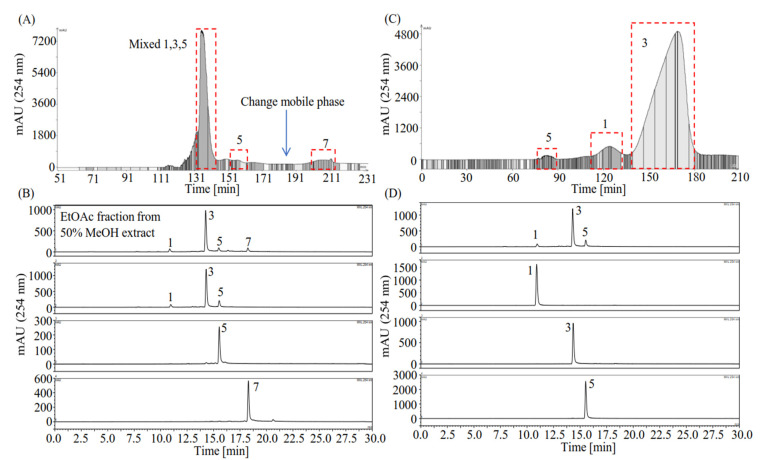
HSCCC and HPLC chromatograms of the EtOAc fraction of the 50% MeOH extract of *Lepechinia meyenii* (Walp.) Epling (*L. meyenii*). (**A**) Polarity-gradient HSCCC separation of the EtOAc fraction of the 50% MeOH extract of *L. meyenii* using the upper layer (PU) of *n*-hexane/EtOAc/MeOH/water (2:5:2:5, *v*/*v*) as the stationary phase, and 10% volume MeOH-modified lower layer (PL + 10% MeOH, *v*/*v*) of the solvent system as the first mobile phase to elute compounds **1**, **3**, and **5**, and 40% volume MeOH-modified lower layer (PL + 40% MeOH, *v*/*v*) of the solvent system as the second mobile phase for isolating compound **7**. The elution rate was 3 mL/min. (**B**) HPLC chromatograms of the EtOAc fraction and the isolated compounds by polarity-gradient HSCCC. (**C**) HSCCC separation of compounds **1**, **3**, and **5** from the polarity-gradient HSCCC subfraction, using PL + 10% MeOH (*v*/*v*) as the stationary phase, and PU as the mobile phase with an elution rate of 3 mL/min. (**D**) HPLC chromatograms of the polarity-gradient HSCCC subfraction and the separated compounds **1**, **3**, and **5** by the second time HSCCC. Notably, the compounds to be separated were later identified as caffeic acid (**1**), rosmarinic acid (**3**), methyl rosmarinate (**5**), and butyl rosmarinate (**7**).

**Figure 4 plants-10-02773-f004:**
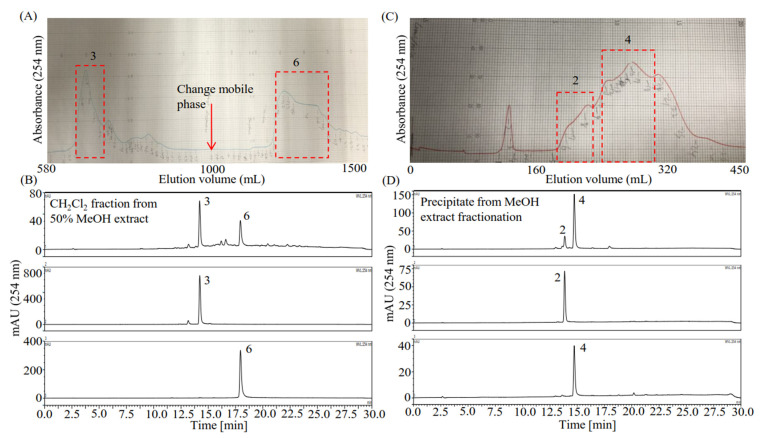
Pre-HPLC separation of the compounds from the H_2_Cl_2_ fraction of the 50% MeOH extract and the partition precipitate from the MeOH extract fractionation process. (**A**) Pre-HPLC separation of components **3** and **6** from the H_2_Cl_2_ fraction of the 50% MeOH extract. The sample (100 mg) was first eluted by 50% MeOH (0–1000 mL; 4 mL/min) to obtain compound **3** (20.6 mg) and then eluted by 60% MeOH (1000–1600 mL; 4 mL/min) to obtain compound **6** (16 mg). (**B**) HPLC profiles of the CH_2_Cl_2_ fraction of 50% MeOH extract and the separated compounds **3** and **6**. (**C**) Pre-HPLC separation of components **2** and **4** from the partition precipitate. Compound **2** (8.8 mg) and compound **4** (6.9 mg) were separated from the mixture (37.6 mg) using 65% MeOH (0–450 mL; 4 mL/min). (**D**) HPLC profiles of the partition precipitate of the MeOH extract and the separated compounds **2** and **4**. Notably, the compounds to be separated were later identified as hesperidin (**2**), rosmarinic acid (**3**), diosmin (**4**), and diosmetin (**6**).

**Figure 5 plants-10-02773-f005:**
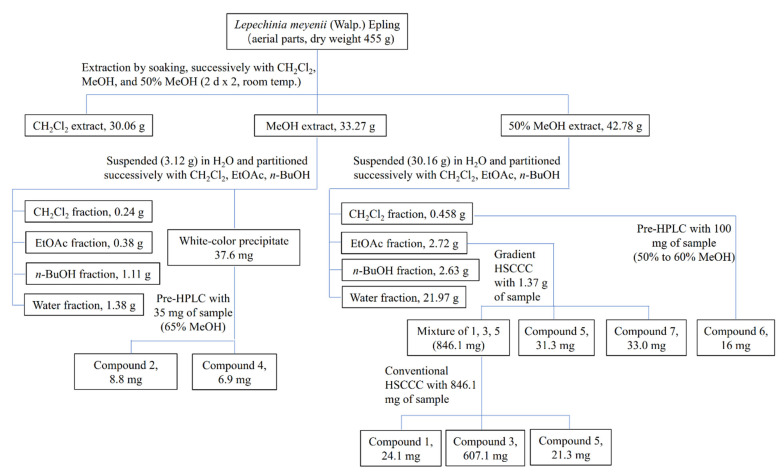
Extraction, fractionation, and separation processes of *Lepechinia meyenii* (Walp.) Epling (*L. meyenii*). Notably, compounds **1**–**7** were later identified as caffeic acid (**1**), hesperidin (**2**), rosmarinic acid (**3**), diosmin (**4**), methyl rosmarinate (**5**), diosmetin (**6**), and butyl rosmarinate (**7**).

**Figure 6 plants-10-02773-f006:**
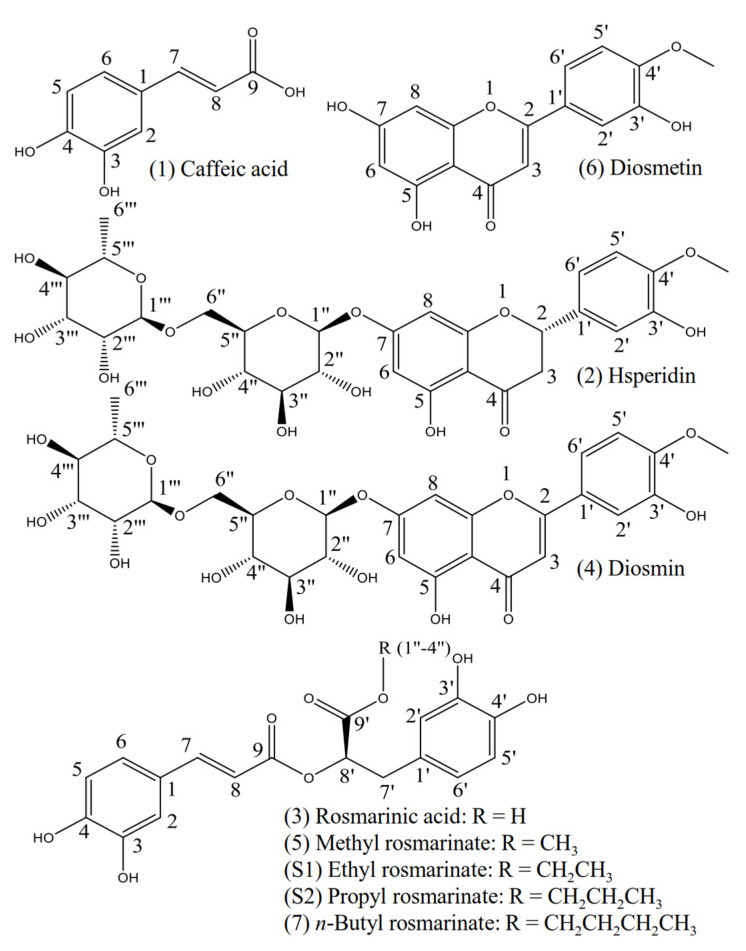
Structures of the separated compounds and synthetic compounds. Compounds **1**–**7** were separated from *Lepechinia meyenii* (Walp.) Epling, whereas **S1** and **S2** were synthetic components.

**Figure 7 plants-10-02773-f007:**
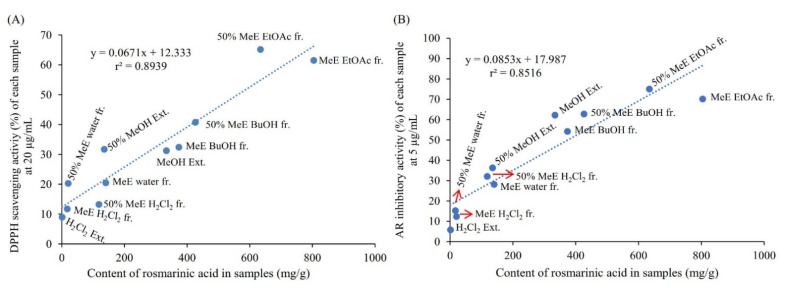
Correlations of the content of rosmarinic acid in the extracts/fractions of *Lepechinia meyenii* (Walp.) Epling and their activities against DPPH radicals (sample concentration 20 µg/mL) (*r* = 0.945, *p* < 0.001) (**A**) and aldose reductase (AR) (sample concentration 5 µg/mL) (*r* = 0.923, *p* < 0.001) (**B**). “MeE”, “50% MeE”, “Ext.” and “fr.” are the abbreviations of “MeOH extract”, “50% MeOH extract”, “extract” and “fraction”, respectively.

**Table 1 plants-10-02773-t001:** Antioxidant and aldose reductase (AR) inhibitory activity of the extracts and fractions of *L. meyenii*.

Sample	DPPH Scavenging Activity			AR Inhibitory Activity (%)		
	Concentration (µg/mL)	Inhibition (%)	IC_50_ (µg/mL)	Concentration (µg/mL)	Inhibition (%)	IC_50_ (µg/mL)
CH_2_Cl_2_ extract	40	16.54 ± 1.16	-	10	11.08 ± 0.62	-
	20	8.92 ± 1.00		5	7.81 ± 0.40	
	10	2.80 ± 0.13		2.5	5.79 ± 0.53	
MeOH extract	40	60.61 ± 3.42	32.81 ± 1.35 ^ab^	2.5	62.16 ± 2.38	1.64 ± 0.10 ^e^
	20	31.22 ± 0.89		1.25	41.67 ± 1.32	
	10	16.46 ± 0.55		0.625	23.48 ± 0.22	
MeE CH_2_Cl_2_ fr.	40	19.78 ± 0.69	-	10	57.49 ± 4.88	8.34 ± 0.94 ^a^
	20	11.73 ± 0.33		5	30.06 ± 0.66	
	10	5.91 ± 0.76		2.5	15.26 ± 1.32	
MeE EtOAc fr.	20	61.44 ± 0.49	15.48 ± 0.04 ^d^	2.5	70.09 ± 1.55	1.24 ± 0.06 ^e^
	10	37.60 ± 0.89		1.25	50.50 ± 0.45	
	5	18.31 ± 0.57		0.625	30.15 ± 1.98	
MeE BuOH fr.	40	62.80 ± 0.92	31.54 ± 1.00 ^b^	5	68.80 ± 0.45	1.94 ± 0.24 ^d^
	20	32.38 ± 1.48		2.5	54.21 ± 0.23	
	10	17.72 ± 0.28		1.25	41.67 ± 4.74	
MeE water fr.	40	36.49 ± 0.55	-	10	68.57 ± 5.14	5.35 ± 0.49 ^b^
	20	20.43 ± 0.84		5	48.39 ± 1.98	
	10	10.52 ± 0.37		2.5	28.13 ± 0.45	
50% MeOH extract	40	57.89 ± 1.75	34.04 ± 0.89 ^a^	5	56.32 ± 0.93	4.02 ± 0.17 ^c^
	20	31.67 ± 0.84		2.5	36.26 ± 1.54	
	10	15.08 ± 1.06		1.25	20.73 ± 1.14	
50% MeE CH_2_Cl_2_ fr.	40	26.51 ± 1.02	-	10	67.57 ± 0.81	5.05 ± 0.12 ^bc^
	20	13.23 ± 1.09		5	49.59 ± 0.83	
	10	7.94 ± 0.53		2.5	32.05 ± 1.7	
50% MeE EtOAc fr.	20	65.12 ± 0.90	14.81 ± 0.14 ^d^	2.5	75.03 ± 0.88	0.86 ± 0.07 ^e^
	10	37.18 ± 0.38		1.25	58.77 ± 1.32	
	5	18.73 ± 0.89		0.625	42.28 ± 2.41	
50% MeE BuOH fr.	40	74.23 ± 3.16	26.20 ± 0.70 ^c^	2.5	62.72 ± 1.32	1.23 ± 0.02 ^e^
	20	40.76 ± 0.46		1.25	53.16 ± 0.55	
	10	21.16 ± 0.16		0.625	35.00 ± 0.66	
50% MeE water fr.	40	36.57 ± 0.65	-	10	47.81 ± 2.01	-
	20	20.26 ± 0.56		5	24.65 ± 0.46	
	10	8.91 ± 0.75		2.5	12.34 ± 0.45	
Quercetin	20	84.78 ± 2.36	10.46 ± 0.34 ^e^	10	64.04 ± 0.88	4.34 ± 0.06 ^bc^
	10	54.43 ± 2.07		5	55.26 ± 1.58	
	5	28.65 ± 0.38		2.5	38.74 ± 0.91	

Note: “MeE”, “50% MeE”, and “fr.” are the abbreviations of “MeOH extract”, “50% MeOH extract”, and “fraction”, respectively. Quercetin was used as a positive control. Different superscript letters (^a,b,c,d,e^) in each IC_50_ column indicate significant differences (*p* < 0.05), and “-” means the IC_50_ values were not available within the concentrations tested.

**Table 2 plants-10-02773-t002:** Screening of the HSCCC solvent system.

Solvent Systems *n*-Hexane/EtOAc/MeOH/Water (*v*/*v*)	*K* Values of Compounds 1, 3, 5, 7				*α_K_* _1/*K*3_
	1	3	5	7	
4:5:4:5	0.12	0.06	0.15	0.68	2.00
3:5:3:5	0.37	0.32	0.67	2.64	1.16
2:5:2:5	1.16	0.90	2.59	7.89	1.29
1:5:1:5	4.94	3.45	10.10	56.98	1.43
2:5:2:5 + 0.1% acetic acid	1.14	0.89	2.62	8.66	1.27

**Table 3 plants-10-02773-t003:** Modification of the solvent system *n*-hexane/EtOAc/MeOH/water (2:5:2:5, *v*/*v*) by adding MeOH.

Addition of MeOH to PL ^1^ (*v*/*v*)	*K*^2^ Values of Compounds 1, 3, 5, 7				*α_K1_* _/*K*3_
	1	3	5	7	
PL alone	1.16	0.90	2.59	7.89	1.29
PL + 10% MeOH ^3^	0.85	0.58	1.42	21.15	1.47
PL + 20% MeOH ^4^	0.6	0.37	1.22	6.20	1.62
PL + 40% MeOH ^5^	0.23	0.10	0.32	0.67	2.30

^1^ Partitioned lower layer (PL) of the solvent system *n*-hexane/EtOAc/MeOH/water (2:5:2:5, *v*/*v*). ^2^
*K* values were obtained by the new solvent system paired by the partitioned upper layer of *n*-hexane/EtOAc/MeOH/water (2:5:2:5, *v*/*v*) and the MeOH-modified PL. ^3^ An extra 10% volume of MeOH was added to PL for polarity modification. ^4^ An extra 20% volume of MeOH was added to PL for polarity modification. ^5^ An extra 40% volume of MeOH was added to PL for polarity modification.

**Table 4 plants-10-02773-t004:** Antioxidant and aldose reductase (AR) inhibitory activity of the separated compounds from *L. meyenii* and synthetic ethyl and propyl rosmarinates.

Sample	DPPH Scavenging Activity			AR Inhibitory Activity (%)		
	Concentration (µM)	Inhibition (%)	IC_50_ (µM)	Concentration (µM)	Inhibition (%)	IC_50_ (µM)
Caffeic acid (**1**)	50	43.33 ± 0.69	-	50	20.34 ± 0.62	-
	25	20.21 ± 0.58		25	17.76 ± 0.41	
	12.5	8.60 ± 0.87		12.5	12.31 ± 2.09	
Hesperidin (**2**)	50	3.37 ± 1.32	-	50	11.63 ± 0.41	-
	25	2.57 ± 0.77		25	10.00 ± 1.22	
	12.5	1.20 ± 1.18		12.5	1.78 ± 0.33	
Rosmarinic acid (**3**)	50	67.04 ± 0.64	36.91 ± 0.35 ^a^	12.5	75.97 ± 1.03	4.08 ± 0.11 ^b^
	25	34.35 ± 0.69		6.25	62.93 ± 2.10	
	12.5	19.05 ± 1.33		3.125	42.65 ± 1.63	
Diosmin (**4**)	50	2.61 ± 0.67	-	50	12.31 ± 0.62	-
	25	0.61 ± 1.08		25	11.22 ± 0.82	
	12.5	0.31 ± 0.69		12.5	8.91 ± 0.62	
Methyl rosmarinate (**5**)	50	76.21 ± 1.04	33.01 ± 0.27 ^b^	1.56	56.67 ± 0.85	1.17 ± 0.08 ^c^
	25	37.59 ± 0.60		0.78	40.88 ± 3.47	
	12.5	18.68 ± 0.63		0.39	22.11 ± 0.85	
Diosmetin (**6**)	50	3.18 ± 0.42	-	50	33.95 ± 1.03	-
	25	0.46 ± 0.35		25	21.97 ± 1.25	
	12.5	0.14 ± 0.34		12.5	10.54 ± 3.79	
*n*-Butyl rosmarinate (**7**)	50	84.85 ± 0.81	30.02 ± 0.10 ^c^	1.56	51.36 ± 0.94	1.54 ± 0.04 ^c^
	25	40.73 ± 1.21		0.78	28.64 ± 0.47	
	12.5	19.98 ± 0.29		0.39	13.13 ± 0.47	
Ethyl rosmarinate (**S1**)	50	83.09 ± 0.21	30.54 ± 0.13 ^c^	1.56	60.48 ± 2.05	1.02 ± 0.07 ^c^
	25	38.38 ± 0.53		0.78	43.20 ± 0.62	
	12.5	22.02 ± 0.35		0.39	27.41 ± 2.66	
Propyl rosmarinate (**S2**)	50	77.82 ± 0.46	32.70 ± 0.08 ^b^	1.56	53.67 ± 0.41	1.29 ± 0.01 ^c^
	25	37.15 ± 0.29		0.78	37.89 ± 1.25	
	12.5	17.68 ± 0.83		0.39	12.31 ± 1.84	
Quercetin	50	72.32 ± 0.42	33.19 ± 0.31 ^b^	25	65.10 ± 1.47	16.16 ± 0.73 ^a^
	25	40.39 ± 0.82		12.5	44.97 ± 1.65	
	12.5	22.31 ± 0.92		6.25	31.50 ± 1.70	

Note: Compounds **1**–**7** were separated from *L. meyenii*, but **S1** and **S2** were synthetic components. Quercetin was used as a positive control. Different superscript letters (^a,b,c^) in each IC_50_ column indicate significant differences (*p* < 0.05), and “-” means IC_50_ values were not available within the concentrations tested.

**Table 5 plants-10-02773-t005:** Contents of rosmarinic acid in extracts, fractions, and raw material of *L. meyenii* (mg/g).

Sample	Content (mg Component/g Sample)
H_2_Cl_2_ extract	1.24 ± 1.00
MeOH extract	333.84 ± 2.74
MeE CH_2_Cl_2_ fr.	16.97 ± 1.25
MeE EtOAc fr.	804.07 ± 4.07
MeE BuOH fr.	373.71 ± 2.78
MeE water fr.	140.14 ± 0.11
50% MeOH extract	135.41 ± 0.54
50% MeE CH_2_Cl_2_ fr.	118.54 ± 0.38
50% MeE EtOAc fr.	634.22 ± 3.37
50% MeE BuOH fr.	426.22 ± 2.56
50% MeE water fr.	20.49 ± 0.55
Dried raw material	37.22

Note: “MeE”, “50% MeE”, and “fr.” are the abbreviations of MeOH extract, 50% MeOH extract, and fraction, respectively. Values are presented as mean ± standard deviation. The contents of rosmarinic acid in the extracts and fractions of *L. meyenii* were quantified by HPLC, whereas the content of rosmarinic acid in the dried raw material (aerial parts) of *L. meyenii* was calculated.

## Data Availability

Data is contained within the article or Appendix A.

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
