# Peer review of "Separation and Identification of Antioxidants and Aldose Reductase Inhibitors in Lepechinia meyenii (Walp.) Epling"

_plants, 2021, doi:10.3390/plants10122773_

Round 1

Reviewer 1 Report

The manuscript entitled "Separation and Identification of Antioxidants and Aldose Reductase Inhibitors in Lepechinia meyenii (Walp.) Epling" presents results dealing with the use of several chromatographic and spectroscopic techniques for the isolation and characterization of some phytochemicals from Lepechinia meyenii. The further antioxidant and AR inhibition activities have been also explored. The manuscript is interesting and well written. I advice the authors to take into account the following comments:

  • In the introduction section please clearly indicated what have been done by the same authors (in reference 15) and what will be reported in the present work in order to show the originality of this work
  • Concerning the identification of the indicated compounds especially compound 2 and 4, I think that they could not be unambiguously characterized through the indicated experimental data. I think that the linkage of the sugar moiety to the aglycon in addition to the sugars interlinkage could not be univocally determined from the given data. In the section 4.8. Structure Identification, the authors indicated that compounds identification has also been achieved through "comparison with references". In this case please indicated the used references compounds and from where these have been purchased.
  • Almost all the identified compounds are not new and their spectral data could be found in previously reported data. I think then that it is not necessary to include the obtained data and indicate only the corresponding references where the same data have been obtained.

Author Response

Overall comment: The manuscript entitled "Separation and Identification of Antioxidants and Aldose Reductase Inhibitors in Lepechinia meyenii (Walp.) Epling" presents results dealing with the use of several chromatographic and spectroscopic techniques for the isolation and characterization of some phytochemicals from Lepechinia meyenii. The further antioxidant and AR inhibition activities have been also explored. The manuscript is interesting and well written. I advice the authors to take into account the following comments:

Response: thank you very much for your positive comments and kind suggestions, we have further improved the manuscript as you suggested.

Specific comments,

Comment 1: In the introduction section please clearly indicated what have been done by the same authors (in reference 15) and what will be reported in the present work in order to show the originality of this work

 Response: thank you very much for your good suggestion. to clearly indicate what have been done by reference 15, the sentence was re-written as “we found that the 70% MeOH extract of L. meyenii has strong antioxidant and AR inhibitory activities” in line 59. Moreover, the sentence in lines 65 and 66 was re-written as “Nevertheless, the antioxidant and AR inhibitory compounds in L. meyenii remain unidentified, which prompted us to separate and identify the underlying bioactive compounds from this plant” to show the originality of this work.

Comment 2: Concerning the identification of the indicated compounds especially compound 2 and 4, I think that they could not be unambiguously characterized through the indicated experimental data. I think that the linkage of the sugar moiety to the aglycon in addition to the sugars interlinkage could not be univocally determined from the given data. In the section 4.8. Structure Identification, the authors indicated that compounds identification has also been achieved through "comparison with references". In this case please indicated the used references compounds and from where these have been purchased

Response: thank you very much for your professional comments. Indeed, the assignment labels of the severely overlapped sugar protons between 3.0 ppm and 3.5 ppm for compounds 2 and 4 were very challenging and that’s why structural elucidation of hesperidin (2) and diosmin (4) using 1H NMR normally only report the assignment labels for glucosyl H-1" and rhamnosyl H-1"' and H-6'" in most of the published papers. In this study, the sequence-specific assignment labels of the sugar protons were carried out by assessing the 600 MHz 1H-1H COSY NMR data. However, as you pointed out, the assignment labels of the sugar protons are not absolutely unambiguous. Therefore, we emphasized that “Please note that the assignment labels of the severely overlapped sugar protons between 3.0 ppm and 3.5 ppm for compounds 2 and 4 were tentatively assigned according to their 1H-1H COSY NMR data” in the footnotes of Table S2, Figure S4.2, Figure S4.4, Figure S4.5, Figure S4.6, Figure S5.4, Figure S5.5, Figure S5.6, and Figure S5.7 in the supplementary data.

Moreover, the reference compound rosmarinic acid (97%, Sigma) was indicated in lines 447 and 592.

Comment 3: Almost all the identified compounds are not new and their spectral data could be found in previously reported data. I think then that it is not necessary to include the obtained data and indicate only the corresponding references where the same data have been obtained.

Response: thank you very much for your suggestion. We have moved all the NMR data to the supplementary data to keep structural evidence (Table S1 and Table S2). However, we would like to delete the NMR data if you insist on deleting them.

Reviewer 2 Report

The manuscript is complex, complete, well done and with many important results, which justifies its publication. 

Author Response

Overall comment: The manuscript is complex, complete, well done and with many important results, which justifies its publication.

Response: thank you very much for your positive comments, we further revised the manuscript to improve its quality.

Reviewer 3 Report

The study was very well designed and presented. The manuscript is clear, well written, and with appropriate conclusions.

There are some minor corrections suggested from my side.

In the Abstract, I feel that the sentence " Particularly, rosmarinic acid is the key contributor...." is too long and complicated. It should be rephrazed and shortened.

In the last sentence in the Introduction "aid" should be replaced with "acid".

In Fig. 5 for the quantity of compound 5, space is placed between "m g".

Regarding the results of the antioxidant activity ann AR inhibition, is there any other reference standard that can be used besides quercetin?

Also, generally, there is no need to present the results for all applied concentrations, the IC50 value is enough. Where IC50 cannot be calculated you may leave only the percent for the highest concentration applied.

In the Discussion, you stated that this plant is richer in rosmarinic acid than many others, but it certainly may depend on the habitat and growing conditions.

Rosmarinus officinalis should be written with the initial capital letter.

Author Response

Overall comment: The study was very well designed and presented. The manuscript is clear, well written, and with appropriate conclusions. There are some minor corrections suggested from my side.

Response: thank you very much for your positive comments and kind suggestions, we have further improved the manuscript as you suggested.

Specific comments,

Comment 1: I feel that the sentence " Particularly, rosmarinic acid is the key contributor...." is too long and complicated. It should be rephrazed and shortened.

 Response: thank you very much for your suggestion. this sentence was rephrased to two sentences as shown in line 28.

Comment 2: In the last sentence in the Introduction "aid" should be replaced with "acid".

 Response: thank you so much for your careful checking. The “aid” was corrected to “acid” in line 79.

Comment 3: In Fig. 5 for the quantity of compound 5, space is placed between "m g".

 Response: thank you very much for your careful checking. The quantity of compound 5 in Fig. 5 was corrected to “mg”

Comment 4: Regarding the results of the antioxidant activity and AR inhibition, is there any other reference standard that can be used besides quercetin?

 Response: thank you very much for your question. Besides quercetin, quercitrin and epalrestat (one proved AR inhibitor drug) are also reference standards in AR inhibition assay. Quercitrin and epalrestat are more active than quercetin against AR, however, quercitrin and epalrestat are very expensive and, therefore, quercitrin and epalrestat are not very common reference standards. In contrast, quercetin is one popular reference standard used in AR inhibition assay. Nevertheless, we have previously compared these three compounds regarding their AR inhibitory activity (please see https://doi.org/10.3390/foods10051079).  

Comment 5: Also, generally, there is no need to present the results for all applied concentrations, the IC50 value is enough. Where IC50 cannot be calculated you may leave only the percent for the highest concentration applied.

 Response: thank you very much for your suggestion. we fully agree with you about only listing the IC50 values. However, considering the activity results in Table 4 were further used to assess the correlations of the content of rosmarinic acid in the extracts/fractions of L meyenii and their activities against DPPH radical and AR in Figure 7, we would like to keep the presented results for all the applied concentrations. However, we would like to delete the data if you insist on deleting them.

Comment 6: In the Discussion, you stated that this plant is richer in rosmarinic acid than many others, but it certainly may depend on the habitat and growing conditions.

 Response: thank you very much for your professional comment. For a more accurate presentation of the result, “higher than in” was replaced with “comparable to” in line 371.

Comment 7: Rosmarinus officinalis should be written with the initial capital letter.

 Response: thank you very much for your careful checking. Rosmarinus officinalis was written with the initial capital letter in line 372.
